# Short-term depression and long-term plasticity together tune sensitive range of synaptic plasticity

**Nicolas Deperrois** ◉ ¤, **Michael Graupner** ◉ *

Université de Paris, CNRS, SPPIN - Saints-Pères Paris Institute for the Neurosciences, F-75006 Paris, France

¤ Current address: Universität Bern, Institut für Physiologie, Bühlplatz 5, CH-3012 Bern, Switzerland
* michael.graupner@parisdescartes.fr

**Data Availability Statement:** All python scripts used to run the simulations presented here are available from the GitHub repository: https://github.com/mgraupe/CalciumBasedPlasticityModel.

## Abstract

Synaptic efficacy is subjected to activity-dependent changes on short- and long time scales. While short-term changes decay over minutes, long-term modifications last from hours up to a lifetime and are thought to constitute the basis of learning and memory. Both plasticity mechanisms have been studied extensively but how their interaction shapes synaptic dynamics is little known. To investigate how both short- and long-term plasticity together control the induction of synaptic depression and potentiation, we used numerical simulations and mathematical analysis of a calcium-based model, where pre- and postsynaptic activity induces calcium transients driving synaptic long-term plasticity. We found that the model implementing known synaptic short-term dynamics in the calcium transients can be successfully fitted to long-term plasticity data obtained in visual- and somatosensory cortex. Interestingly, the impact of spike-timing and firing rate changes on plasticity occurs in the prevalent firing rate range, which is different in both cortical areas considered here. Our findings suggest that short- and long-term plasticity are together tuned to adapt plasticity to area-specific activity statistics such as firing rates.

## Author summary

Synaptic long-term plasticity, the long-lasting change in efficacy of connections between neurons, is believed to underlie learning and memory. Synapses furthermore change their efficacy reversibly in an activity-dependent manner on the subsecond time scale, referred to as short-term plasticity. It is not known how both synaptic plasticity mechanisms—long- and short-term—interact during activity epochs. To address this question, we used a biologically-inspired plasticity model in which calcium drives changes in synaptic efficacy. We applied the model to plasticity data from visual- and somatosensory cortex and found that synaptic changes occur in very different firing rate ranges, which correspond to the prevalent firing rates in both structures. Our results suggest that short- and long-term plasticity act in a well concerted fashion.

**Funding:** MG was supported by the French Agence Nationale de la Recherche (ANR, WalkingCrossingNeurons, ANR-18-CE37-0006-01). The funders had no role in study design, data collection and analysis, decision to publish, or preparation of the manuscript.

**Competing interests:** The authors have declared that no competing interests exist.

## Introduction

The impact of presynaptic action potentials on the postsynaptic neuron's excitability varies on multiple time-scales; successive presynaptic spikes produce short-term depression or facilitation lasting for a few minutes, while prolonged pre- and postsynaptic stimulation induce long-term plasticity. How the two interact during activity epochs remains little studied.

Experimental studies have shown that the induction of synaptic long-term potentiation (LTP) and depression (LTD) depends on (i) the firing rates of pre- and postsynaptic neurons [1, 2] and on (ii) the precise timing of pre- and postsynaptic action potentials [3–6]. Studies in different brain regions have revealed marked differences in the dependence of plasticity on spike-timing and frequency [7]. Furthermore, electrical postsynaptic responses increase and/or decrease upon presynaptic stimulation in a history-dependent manner known as short-term plasticity (STP). A dynamic enhancement of the postsynaptic reponse is termed short-term facilitation and the reduction is called short-term depression (STD). While short-term facilitation has been attributed to the influx of calcium in the presynaptic terminal, short-term depression is attributed to the depletion of some pool of presynaptic vesicles [8, 9]. Different synapses exhibit varied forms of short-term plasticity, being either depression dominated, faciliation dominated, or a mixture of both [10–12]. Activity-dependent depression dominates synaptic transmission between neocortical pyramidal neurons [13]. STP has been proposed as a mechanism serving as dynamic filter for information transmission (see [14]). However, the role of short-term synaptic changes for long-term plasticity induction has attracted little attention.

Plasticity models of different complexities and degrees of biological realism have been developed to capture the link between one or several stimulation protocols that induce long-term plasticity. The classical spike-timing based models capture pair-based spike-time dependent plasticity (STDP; [15–17]), but do not account for the firing rate-dependence of plasticity (but see [18]). To account for the firing-rate dependence and non-linearities of plasticity induction, more complex models have been proposed following phenomenological and biophysical directions (see [19] and [20]). We here focus on a biophysically inspired model in which both pre- and postsynaptic spike-trains induce postsynaptic calcium elevations which in turn drive plastic changes [21, 22] as this type of model lends itself easily to incorporate STP. Similarly, short-term plasticity models have been proposed based on the vesicle depletion model ([23]; see [24]). This model was later extended by release probability facilitation to account for the mixture of both facilitation and depression present at some synapses [10, 25–27].

Short-term synaptic plasticity has been considered in a calcium-dependent model of long-term plasticity and applied to describe plasticity induced by spike-triplets [28, 29]. In this model and similar to suppression models [30, 31], multiple spikes originating from the same neuron are subject to short-term depression reducing both pre- and postsynaptic spike efficacies which improves the match with the experimental plasticity data for complex spike patters [28–31]. In these suppression models, the interplay between spike-timing and firing rate induction protocols has not been systematically studied. In a different phenomenological approach, Costa et al. [32] account for the change of short-term plasticity after the induction of long-term plasticity. However, the induction of long-term changes is not subjected to short-term dynamics in that model. Furthermore, the combination of short- and long-term plasticity has been suggested to limit the boundless growth of synapses without weight constraints in a spike-timing based model [33].

How do short- and long-term plasticity interact during induction protocols of synaptic long-term changes using combinations of firing rate and spike-timing? Are the brain region and synapse specific realizations of short- and long-term plasticity mechanisms related? These

questions are pertinent since long-term changes are driven by calcium and calcium transients are affected by short- and long-term plasticity dynamics. To address this, we integrate a deterministic model of short-term depression [34–36] in the calcium-based model of long-term plasticity [21, 22, 28, 29]. The short-term dynamics parameters account for electrical responses recorded between layer V pyramidal neurons in visual- and somatosensory cortex and are applied to the calcium transients. We fit the calcium-based model with STP to synaptic long-term plasticity data obtained in both brain-regions and we quantify the influence of spike-timing and firing rate changes on the plasticity. To simulate changes in synaptic efficacy under more natural conditions, we use irregular spike patterns [22]. We find that short- and long-term plasticity are tuned together such that the sensitive range of synaptic plasticity is located at firing rate ranges, which match the prevalent firing rates in the respective cortical regions.

## Materials and methods

### Calcium-based model of synaptic plasticity

We investigate the calcium-based model where the postsynaptic calcium concentration determines the temporal evolution of the synaptic weight variable, $w$. The postsynaptic calcium in turn is as a function of pre- and postsynaptic activity and depends on the current synaptic weight. The model is studied extensively in [21] and [22].

Shortly, the postsynaptic calcium concentration drives changes in $w$ according to

$$\tau \dot{w} = \gamma_p (1 - w) \Theta[c(t) - \theta_p] - \gamma_d w \Theta[c(t) - \theta_d]. \tag{1}$$

$\tau$ is the time constant of synaptic efficacy changes happening on the order of seconds to minutes. The two terms on the right-hand-side of Eq (1) describe in a highly simplified fashion calcium-dependent signaling cascades leading to synaptic potentiation and depression, respectively. The synaptic efficacy variable tends to increase, or decrease, when the instantaneous calcium concentration, $c(t)$, is above the potentiation $\theta_p$ or the depression threshold $\theta_d$, respectively ($\Theta$ denotes the Heaviside function, $\Theta[c - \theta] = 0$ for $c < \theta$ and $\Theta[c - \theta] = 1$ for $c \geq \theta$). The parameter $\gamma_p$ (resp. $\gamma_d$) measures the rate of synaptic increase (resp. decrease) when the potentiation (resp. depression) threshold is exceeded.

Here, we consider the evolution of a population of synapses. $w$ therefore describes the *mean* synaptic weight dynamics of a number of synapses forming connections between two neurons. In turn, an activity-dependent noise term appearing in earlier versions of the model [21, 22] is not considered. In the absence of activity the synapse has a continuum of stable states in Eq (1). In other words, $w$ is stable at every value [0, 1] for $c < \min(\theta_d, \theta_p)$.

### Calcium dynamics implementing short-term synaptic depression

The postsynaptic calcium dynamics describes the compound calcium trace resulting from pre- and postsynaptic activity. Calcium increases from presynaptic activity are subject to short-term depression and long-term plasticity.

The average dynamics of short-term synaptic depression can be captured by assuming finite presynaptic resources. At the event of a presynaptic action potential, a fraction of the resources is utilized to evoke a postsynaptic response. If a subsequent presynaptic action potential arrives before all the utilized resources are recovered, the following postsynaptic response will be smaller [36].

Calcium transients induced by presynaptic activity depend furthermore on the current synaptic weight. And the calcium dynamics in turn drives long-term synaptic changes. To account for this coupling between postsynaptic calcium dynamics and the synaptic weight, we assume

that the current synaptic weight variable $w$ linearly scales the presynaptically induced calcium amplitude.

Calcium transients evoked by presynaptic spikes, $c_{pre}$, are described by

$$\frac{dx}{dt} = \frac{1-x}{\tau_{rec}} - Ux\sum_i \delta(t - t_i - D),$$ (2)

$$\frac{dc_{pre}}{dt} = -\frac{c_{pre}}{\tau_{Ca}} + wC_{pre}Ux\sum_i \delta(t - t_i - D),$$ (3)

where $x$ denotes the fraction of available presynaptic resources. $U$ determines the fraction of the resources utilized at each presynaptic spike, and $\tau_{rec}$ is the time constant of resource recovery back to the resting state of $x = 1$. $\tau_{Ca}$ is the calcium decay time constant. $C_{pre}$ is the presynaptically evoked calcium amplitude which is scaled by the current value of the synaptic weight, $w$ (Eq (1)). Note that an isolated presynaptic spike induces a maximal calcium transient of amplitude $wC_{pre}U$ (since all presynaptic resources are available, *i.e.*, $x(t) = 1$), and subsequent spikes induce amplitudes of $wC_{pre}Ux$. The sums run over all presynaptic spikes occurring at times $t_i$. The time delay, $D$, between the presynaptic spike and the occurrence of the corresponding postsynaptic calcium transient accounts for the slow rise time of the NMDAR-mediated calcium influx.

The parameters describing short-term depression in somatosensory cortex are $U = 0.46$ and $\tau_{rec} = 525$ ms [36]. Using the same approach as described in [36], we fitted the short-term depression model (Eqs (2) and (3)) to voltage traces recorded between L5 neurons of visual cortex (Fig. 1*C* in [37]). Utilizing a least squares fitting routine, we obtained $U = 0.385$ and $\tau_{rec} = 149$ ms (see Table 1 and Fig 1*A*).

For comparison, we also study the calcium-based plasticity model *without* short-term depression. In this case, the presynaptically evoked calcium amplitude is $wC_{pre}$ in response to

**Table 1. Parameters for short-term and long-term plasticity in visual- and somatosensory cortex.**

| plasticity type | parameter | unit | visual cortex | | | somatosensory cortex | | |
|---|---|---|---|---|---|---|---|---|
| short-term | $U$ | | 0.3838 | - | 0.3838 | 0.46 | - | 0.46 |
| | $\tau_{rec}$ | ms | 148.9192 | - | 148.9192 | 525 | - | 525 |
| long-term | $\tau_{Ca}$ | ms | 38.3492083 | 32.1900754 | 36.1126107 | 48.9774484 | 34.0495917 | 85.8919093 |
| | $C_{pre}$ | | 3.99132241 | 1.60681037 | 0.353083257 | 2.41618557 | 0.5081618 | 0.931917611 |
| | $C_{post}$ | | 1.12940834 | 1.1243642 | 1.46971648 | 1.38836494 | 1.43328377 | 1.24804789 |
| | $n$ | | 1 | 1 | 2 | 1 | 1 | 2 |
| | $\theta_d$ | | 1 | 1 | 1 | 1 | 1 | 1 |
| | $\theta_p$ | | 1.63069609 | **1.63069609** | 2.31445884 | 1.38843434 | **1.38843434** | 1.93270668 |
| | $\gamma_d$ | | 111.320539 | 31.9759883 | 183.511795 | 176.541097 | 105.05417 | 157.338766 |
| | $\gamma_p$ | | 564.392975 | 161.987985 | 1000.000 | 579.578738 | 406.983648 | 518.174280 |
| | $\tau$ | sec | 299.8778 | 79.9756573 | 525.924639 | 143.096290 | 26.5966635 | 196.775963 |
| | $D$ | ms | 9.23545841 | 5.75272377 | 5.51651933 | 10.0700540 | 8.37904652 | 5.0 |
| fit quality | **SSD** | | 0.080002 | 0.0314 | 0.0857 | 0.00839 | 0.000041 | 0.01246 |

The short-term plasticity parameters shown in the first two lines are obtained from fitting the STD model to visual cortex voltage traces (Fig. 1C in [37]), and are taken from [36] for the somatosensory cortex. Long-term plasticity parameters are obtained by fitting the calcium-based model with short-term depression (left columns), without STD (middle columns), and with nonlinear calcium dynamics ($n = 2$, right columns) to visual- [2] and somatosensory cortex plasticity data [3]. Values in bold were not allowed to be optimized by the fitting routine. The last line gives the SSD for all cases providing a measure of the fit quality.

*all* presynaptic spikes and the calcium dynamics simplifies to

$$\frac{dc_{\text{pre}}}{dt} = -\frac{c_{\text{pre}}}{\tau_{\text{Ca}}} + wC_{\text{pre}}\sum_i \delta(t - t_i - D). \tag{4}$$

We examined two variants of the postsynaptic calcium implementation: (1) with linear calcium dynamics (as in [21]) where contributions from presynaptic and postsynaptic spikes add linearly (this variant is used throughout the manuscript); and (2) a nonlinear version of calcium dynamics that accounts for the nonlinear summation of presynaptically and postsynaptically evoked transients when the post-spike occurs after the pre-spike. It describes the nonlinear portion of the NMDAR-mediated calcium current, which is triggered by the coincident occurrence of postsynaptic depolarization from the post-spike and presynaptic activation from the pre-spike. This version was used in Fig 6 and the corresponding paragraph. Below we describe the details of postsynaptically evoked calcium dynamics in the two variants of the implementation.

1. **Linear calcium dynamics**, calcium transients evoked by postsynaptic spikes, $c_{\text{post}}$, are described as

$$\frac{dc_{\text{post}}}{dt} = -c_{\text{post}}/\tau_{Ca} + C_{\text{post}}\sum_j \delta(t - t_j), \tag{5}$$

where $\tau_{Ca}$ is the calcium decay time constant and $C_{\text{post}}$ the postsynaptically evoked calcium amplitude. The sum runs over all postsynaptic spikes occurring at times $t_j$.

2. **In the nonlinear calcium dynamics**, calcium transients evoked by postsynaptic spikes are described as

$$\frac{dc_{\text{post}}}{dt} = -c_{\text{post}}/\tau_{Ca} + C_{\text{post}}\sum_j \delta(t - t_j) + \eta\sum_j \delta(t - t_j)c_{\text{pre}}. \tag{6}$$

$\eta$ implements the increase of the NMDA mediated current in case of coincident presynaptic activation and postsynaptic depolarization. $\eta$ determines by which amount the postsynaptically evoked calcium transient is increased in case of preceding presynaptic stimulation, in which case $c_{\text{pre}} \neq 0$. $\eta$ is related to the experimentally measured nonlinearity factor, $n$, ([38]; peak calcium amplitude normalized to the expected linear sum of pre- and postsynaptically evoked calcium transients) by

$$\eta = \frac{n(C_{\text{post}} + w_0 C_{\text{pre}}U) - C_{\text{post}}}{w_0 C_{\text{pre}}U} - 1, \tag{7}$$

where $w_0 = w(t = 0)$ is the initial value of the synaptic weight in all simulations ($w_0 = 0.5$, see Table 1 for the area specific $U$). We use for the maximal nonlinearity factor $n = 2$ consistent with data from [38].

The *total calcium concentration* is the sum of the pre- and the postsynaptic contributions, $c(t) = c_{\text{pre}}(t) + c_{\text{post}}(t)$. Without loss of generality, we set the resting calcium concentration to zero, and use dimensionless calcium concentrations.

## Fitting the plasticity models to experimental data

In order to compare plasticity between visual- and somatosensory cortex, we fitted three variants of the calcium-based plasticity model to experimental plasticity data obtained from

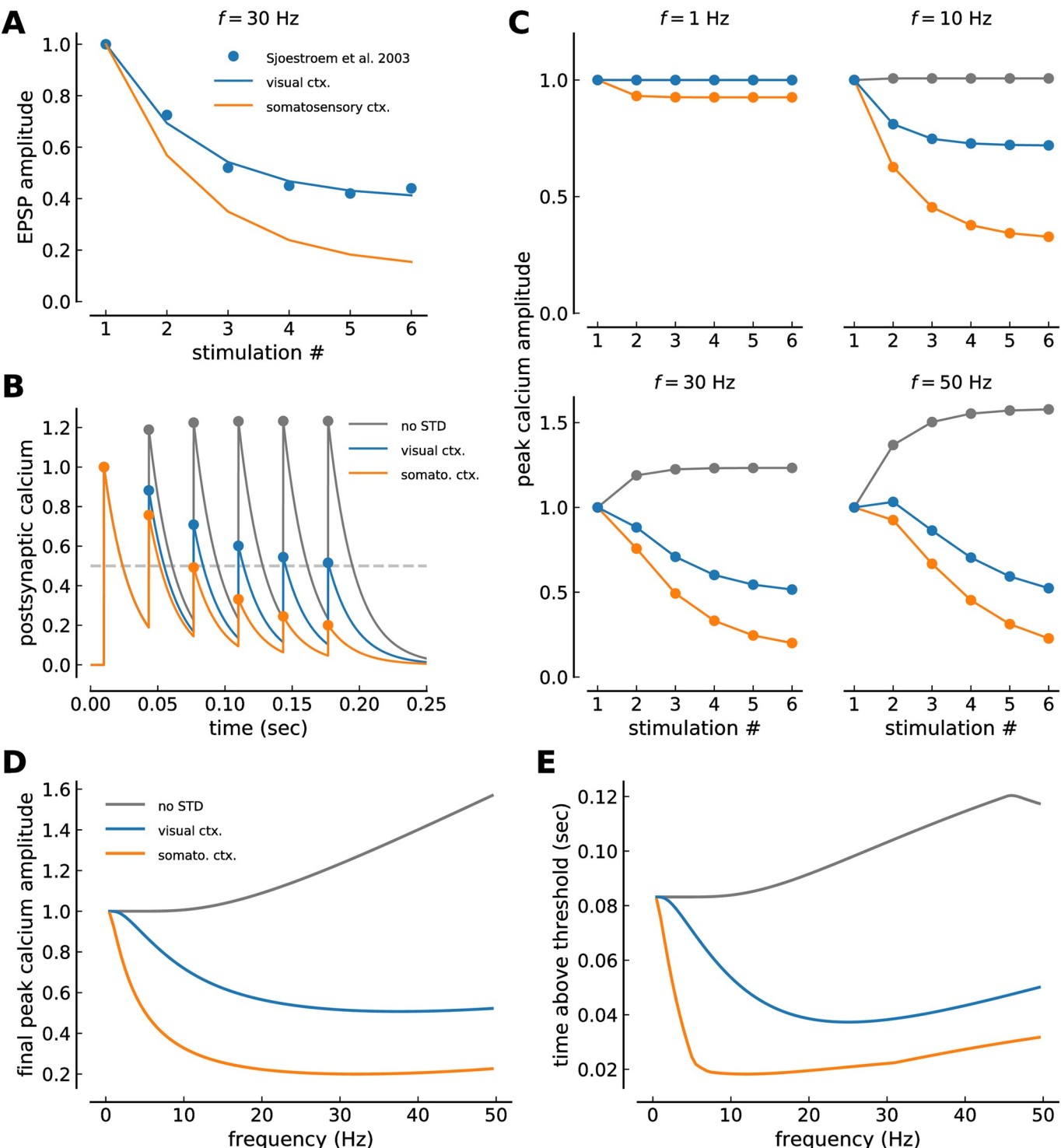

**Fig 1. Calcium dynamics with short-term depression in visual and somatosensory cortex.** *(A)* Short-term depression dynamics of EPSP amplitudes in response to a train of six action potentials at 30 Hz in visual- (blue) and somatosensory cortex (orange). The blue points show data from visual cortex (Fig. 1C in [37]) and the blue line depicts the fit of the short-term depression model (Eqs (2) and (3)). EPSP amplitudes in somatosensory cortex have been fitted using the same STD model [36] and the orange line shows the short-term depression model with the somatosensory parameters (see Materials and methods and Table 1). *(B)* Example simulations of calcium traces with visual cortex STD (blue), somatosensory cortex STD (orange) and no STD (gray line). Traces are generated by six presynaptic stimuli occurring at 30 Hz. Peak calcium amplitudes upon each stimulation are marked by circles. *(C)* Peak calcium amplitudes in response to a train of six stimuli at various frequencies (marked on top of each panel). Peak amplitudes of the calcium trace (see panel *B*) at the time of the stimulation are shown for visual cortex STD (blue), somatosensory cortex STD (orange) and no STD (gray line). *(D)* Final peak calcium amplitude at the sixth presynaptic stimuli as

function of stimulation frequency for visual cortex STD (blue), somatosensory cortex STD (orange) and no STD (gray line). The plotted peak corresponds to the last point in panel C. **(E)** Time spent above threshold by the calcium trace during the train of six presynaptic stimuli as function of stimulation frequency. For illustration, a threshold of $\theta = 0.5$ is used. Color code as in panel D. An initial calcium amplitude of 1 is used for all cases and the calcium decay time constant is taken to be $\tau_{Ca} = 20$ ms in *B-E*. All simulations of the calcium dynamics (panels *B—E*) are based on the calcium-based plasticity model with STD dynamics (Eqs (2) and (3)).

synapses between layer V neurons in the rat visual cortex [2] and between layer V neurons in the rat somatosensory cortex [3]: (1) model with linear calcium dynamics and with the corresponding short-term depression parameters, (2) model with linear calcium dynamics without short-term depression, and (3) model with nonlinear calcium dynamics and short-term depression. Note that the short-term depression parameters were not optimized during the fit to the long-term plasticity data but kept fixed (see previous paragraph, Table 1).

The stimulation protocols employed in visual- [2] and in somatosensory cortex [3] combine spike-timing and firing rate components by varying the presentation frequency of spike-pairs with constant time lag, $\Delta t$. Stimulation patterns are grouped in five pairs of pre- and postsynaptic spikes with long pauses in-between in order to allow for full recovery of short-term depression processes between each group of five pairs. Specifically, [2] repeated the bursts of five pre- and postsynaptic spikes 15 times every 10 s, while [3] repeated the bursts 10 times every 4 s. The time lag between pre- and postsynaptic spikes as well as the presentation frequency of the pairs within the burst were systematically varied in both studies (see experimental data in Fig 2A and 2B).

We consider the change in synaptic strength as the ratio between the synaptic strength after and before the stimulation protocol $w(T)/w_0$, where $T$ marks the end of the stimulation protocol. $w_0 = 0.5$ in all simulations and calculations. As a consequence, the maximally evoked change remains in the interval [0, 2].

We defined the goodness of fit to the experimental plasticity data by a cost function which is the sum of all squared differences (SSD) between data points and the analytical solution for the change in synaptic strength of the calcium-based model. We drew initial parameter values from a uniform distribution and use a downhill Simplex algorithm to search the minimum of the cost function. The fit is repeated $> 10^9$ times and the parameter set with the lowest cost function is used (shown in Table 1). Smooth curves describing spike-timing dependent plasticity with respect to stimulation frequency were imposed (see Fig 2A and 2B).

## Irregular spike-pair stimulation

To study changes in synaptic efficacy induced by firing rate and spike correlations under more natural conditions, we use irregular spike-pair stimulation. This stimulation protocol was proposed and extensively studied in [22]. In short, irregular spike-pairs were generated using discretely correlated Poisson processes. The presynaptic neuron emitted spikes with Poisson statistics at rate $\nu_{pre}$. Each of these presynaptic spikes induced with probability $p$ a postsynaptic spike with a fixed time lag $\Delta t$. The postsynaptic neuron in addition emits independent spikes with Poisson statistics at a rate $\nu_{post} - p\nu_{pre}$, so that the total postsynaptic firing rate is $\nu_{post}$.

We systematically varied the firing rates with $\nu_{pre} = \nu_{post}$, $\Delta t$ and $p$. $p$ effectively controls the strength of spiking correlation between pre- and postsynaptic neuron, with maximal correlation for $p = 1$ (each presynaptic spike is followed by a postsynaptic one) and independent Poisson firing for $p = 0$. The stimulation is imposed for a duration of $T = 10$ s, independently of the firing rate (so that the total number of emitted spikes varies with the firing rate).

To quantitatively compare the influence of firing rates and spike correlations on synaptic changes in irregularly firing neurons, we quantify the sensitivity of synaptic strength to correlations and firing rate changes [22]. These measures give the change in synaptic weight when adding spike correlations to uncorrelated pre- and postsynaptic neurons, or when increasing the firing rate by a certain amount. In short, the synaptic efficacy $w(T)$ at the end of a stimulation protocol of duration $T$ is a random variable, the value of which depends on the precise realization of the pre- and postsynaptic spike trains, their firing rates and their correlation. The average synaptic efficacy $\bar{w}(T)$ can be written as

$$\bar{w}(T) = \bar{w}_{\text{no corr}}(v_{pre}, v_{post}, T) + \bar{w}_{\text{corr}}(v_{pre}, v_{post}, C(t), T), \tag{8}$$

where $\bar{w}_{\text{no corr}}$ is the average synaptic efficacy attained for uncorrelated pre- and postsynaptic spike trains of rates $v_{pre}$ and $v_{post}$. The quantity $\bar{w}_{\text{corr}}$ represents the additional change in synaptic efficacy induced by correlations between the pre- and postsynaptic spike trains. We call $\bar{w}_{\text{corr}}$ the *sensitivity of synaptic strength to correlations*. Note that this sensitivity to correlations depends both on the correlation function $C(t)$ between the pre- and postsynaptic spike trains and individual firing rates of the neurons.

We furthermore quantify the sensitivity to firing rates $\delta\bar{w}_{\text{no corr}}$, defined as the change between the synaptic strength attained at a given, baseline pre- and postsynaptic firing rate, and the synaptic strength attained by increasing the firing rates by a given amount $\delta v$:

$$\delta\bar{w}_{\text{no corr}}(v_{pre}, v_{post}, \delta v) = \bar{w}_{\text{no corr}}(v_{pre} + \delta v, v_{post} + \delta v, T) - \bar{w}_{\text{no corr}}(v_{pre}, v_{post}, T). \tag{9}$$

Note that this sensitivity depends both on the baseline firing rates $v_{pre}$, $v_{post}$, and the amount of increase in the firing rates $\delta v$.

### Event-based implementation

Simulations were performed using an event-based implementation of the calcium-based synaptic plasticity model in an analytically exact way. The synaptic efficacy (Eq (1)), pre- (Eqs (3) or (4)) and postsynaptically evoked calcium transients (Eqs (5) or (6)) are updated only upon the occurrence of pre- and postsynaptic spikes. See [39] for details of the event-based implementation.

For the deterministic, regular stimulation protocols used to fit the model (Fig 2), the spike-pattern of the entire stimulation protocol (see details above) were generated and the synaptic weight was computed once for a given parameter set. For the irregular, Poisson process-based stimulation protocols, spike patterns were drawn randomly for a 10 s long stimulation according to the pre- and postsynaptic firing rates and correlation structure defined by $v_{pre}$, $v_{post}$, $\Delta t$ and $p$. The synaptic weight was initialized at $w_0 = 0.5$ and calculated for $T = 10$ s based on this specific activity pattern. The process was repeated with changing random number seeds and the change in synaptic strength reported for irregular stimulation patterns are averages over 10,000 repetitions.

### Results

To study the interplay between short-term- and long-term plasticity, we used numerical simulations and mathematical analysis of a calcium-based plasticity model [20, 22]. We first extracted synaptic short-term depression dynamics for the respective cortical region from voltage recordings and used these to describe calcium transient dynamics. We then fitted the plasticity model to plasticity results obtained in the visual cortex [2] and the somatosensory cortex [3]. Finally, we studied how firing rate and spike-timing shape plasticity in both cortical areas with irregular activity patterns.

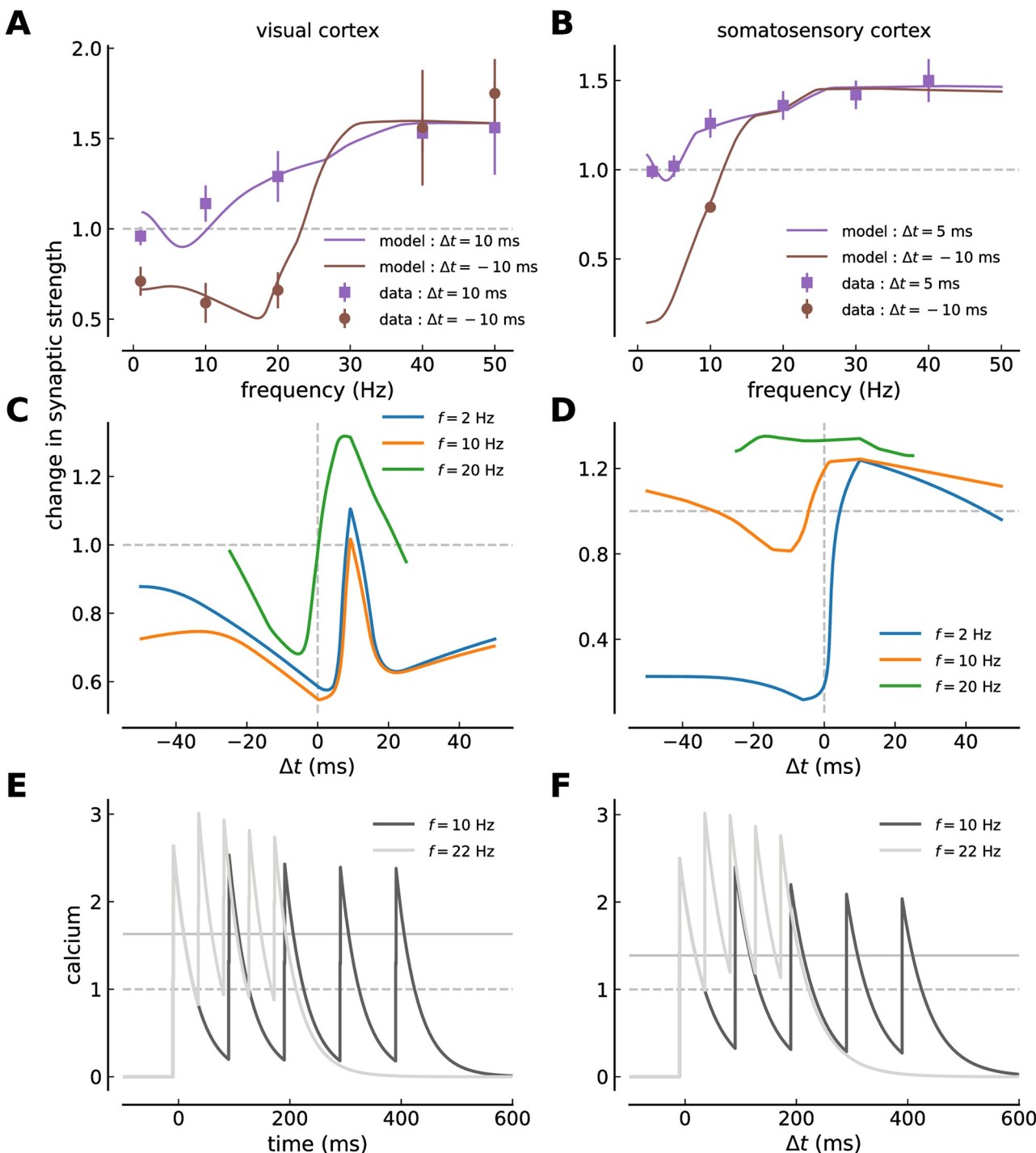

**Fig 2. Calcium-based plasticity model with short-term depression fitted to visual- and somatosensory cortex plasticity data.** *(A, B)* The change of the synaptic strength is shown as a function of the spike-pair presentation frequency for regular pre-post pairs ($\Delta t = 10$ ms in A and $\Delta t = 5$ ms in B, purple squares and lines), and regular post-pre pairs ($\Delta t = -10$ ms in both panels, brown circles and lines). The data-points are taken from plasticity experiments in visual cortex slices (A, mean ± SEM, [2]) and somatosensory cortex slices (B, mean ± SEM, [3]). The solid lines show the model fit to the experimental data. *(C, D)* Change in synaptic strength as a function of the time lag between pre- and postsynaptic spikes for regular spike-pairs at three different spike-pair presentation frequencies (given in panel) in visual- (C) and somatosensory cortex (D). *(E, F)* Calcium traces during regular post-pre spike-pair stimulation for the visual- (E) and somatosensory data-set (F). Five spike-pairs with $\Delta t = -10$ ms are presented at the frequency given in the panel. The full and the dashed gray lines indicate the area-specific thresholds for potentiation and depression, respectively.

## Short-term depression strongly affects the calcium trace during bursts of activity

In order to study the interplay between short-term depression (STD) and long-term plasticity, we first extracted the short-term depression dynamics of postsynaptic responses for visual- and somatosensory cortex. Assuming that the evoked postsynaptic current is directly proportional to the induced calcium concentration, we applied the same STD dynamics to calcium amplitudes elicited by bursts of presynaptic activity.

Short-term depression has been measured and described with regard to the dynamics of the postsynaptic current or voltage response (*e.g.* [36]). STD can be explained by the depletion of neurotransmitters in the presynaptic terminal. Here, short-term depression is modeled as a use-dependent decrease in presynaptic resources [35, 36]. We furthermore assume that evoked postsynaptic current—controlled by the amount of utilized resources, and typically measured in electrophysiological studies—is directly proportional to the induced calcium amplitude and use the description of short-term plasticity dynamics [28, 29, 36] for calcium (Eqs (2) and (3)).

We use a deterministic model of STD which describes the average temporal dynamics of the postsynaptic calcium responses to presynaptic stimulation (see Materials and methods, Eqs (2) and (3)). STD parameters have been characterized for connections between L5 pyramidal neurons in the somatosensory cortex [36]. We fitted the deterministic model to EPSP responses obtained between L5 neurons in visual cortex ([37], Fig 1A, see parameters in Table 1). Comparing both parameter sets reveals differences in STD betweeen visual and somatosensory cortex. More presynaptic resources are utilized in somatosensory cortex upon stimulation (*i.e.*, $U_{\text{vis.ctx.}} < U_{\text{somat.ctx}}$), and the recovery time of these resources is longer in somatosensory cortex (see Table 1 and Fig 1).

STD dynamics applied to calcium drastically alters the postsynaptic calcium response upon repeated presynaptic stimulation. While calcium transients build up and reach a plateau of attained amplitudes without STD (see gray lines in Fig 1C), subsequent calcium transients decrease with STD and this difference increases with frequency (50 Hz case in Fig 1C). Irrespective of responses in visual or somatosensory cortex, the decrease in induced calcium amplitudes due to STD prevents the calcium trace from exceeding the amplitude of the first transient up to high stimulation frequencies ($f < 46$ Hz for visual cortex; $f < 62$ Hz for somatosensory cortex). In turn, no plateau is reached and the transients keep decreasing for consecutive stimulations.

Overall, postsynaptic responses to repeated presynaptic stimulation are suppressed stronger in the somatosensory cortex compared to the visual cortex since the fraction of used resources, $U$, is larger in the somatosensory cortex (Table 1, Fig 1D and 1E). Moreover, the depression happens already at low ($< 5$ Hz) frequencies since the recovery time constant of presynaptic resources is longer in the somatosensory cortex compared to the visual cortex (Fig 1, Table 1). As a consequence, the calcium amplitude (Fig 1D) and the time spent by the calcium trace above a given threshold change drastically in the frequency range up to 5 Hz in the somatosensory cortex, while the change occurs for a larger frequency range in the visual cortex (Fig 1E).

In summary, the differences in STD lead to a strong suppression of postsynaptic responses at low frequencies in the somatosensory cortex while the same suppression occurs over a larger frequency range in the visual cortex.

## Calcium-based model with STD fitted to experimental plasticity data

Plasticity is driven by postsynaptic calcium elevations which is captured by the calcium-based plasticity model, where threshold crossings drive long-term depression and potentiation processes. In turn, the amplitude of presynaptically evoked calcium transients is influenced by

the current synaptic weight creating a coupling between weight and calcium (see Materials and methods). In addition, STD leads to a highly dynamic calcium trace with changing amplitudes upon each stimulation and furthermore prevents considerable build-up of calcium even at relatively high presynaptic stimulation frequencies (Fig 1). Here we ask whether the calcium-based plasticity model with STD can capture the experimental plasticity data obtained by combining spike-timing and frequency stimulation.

Pre- and postsynaptic spikes with delay $\Delta t$ presented in bursts of five pairs at varying frequencies have been shown to induce LTP for pre-post spike-pairs ($\Delta t = 10$ and 5 ms) for frequencies $\geq 10$ Hz in visual- and somatosensory cortices. Post-pre pairs ($\Delta t = -10$ ms) evoke LTD at low frequencies in both structures ($< 30$ Hz) and LTP at high frequencies in the visual cortex (Fig 2A and 2B).

The postsynaptic calcium response to spike-pairs presented in bursts is subjected to STD and the change in synaptic weight. We therefore implemented STD dynamics in the calcium driving plasticity changes in the model, and furthermore assumed that the presynaptically induced calcium amplitude scales linearly with the current synaptic strength. We then fitted the long-term plasticity parameters of the coupled model to plasticity data obtained in visual- and somatosensory cortices. The parameters describing the STD dynamics are specific for each cortical region considered and are kept constant during that fit (see Table 1).

We find that the calcium-based plasticity model with STD of the calcium dynamics captures the experimental data of visual- and somatosensory cortex (Fig 2A and 2B). In particular, the model retains the frequency dependence of plasticity for both, pre-post and post-pre pairs despite the fact that presynaptically evoked calcium amplitudes are subjected to a strong frequency-dependent suppression and a constant drift due to the scaling with the current synaptic weight (which increases—for LTP—or decreases—for LTD—during the stimulation protocol). The fit of the calcium-based model yields STDP curves which are dominated by depression for low pair presentation frequencies (Fig 2C and 2D). Intermediate frequencies yield curves with depression for $\Delta t < 0$ ms and potentiation for $\Delta t > 0$ ms, whereby intermediate frequencies implies $f \approx 20$ Hz in visual cortex and $f \approx 10$ Hz in somatosensory cortex.

Due to STD, we find a non-monotonic behavior of LTD *and* LTP with respect to the stimulation frequency. The model predicts weak LTP for very low presentation frequencies of pre-post ($\Delta t = 10$, 5 ms) pairs (Fig 2A and 2B). This LTP vanishes at frequencies around 5 Hz and re-emerges at higher rates, a behavior which is due to STD-induced reduction in presynaptically evoked calcium and not seen in the model variant without STD (compare with Fig 5A and 5B). Not LTP but no change has been measured in visual cortex at 0.1 Hz where the sparse data points hint to a monotonic increase of LTP (Fig 2A). Such low stimulation frequencies were not investigated in somatosensory cortex (Fig 2B).

One particular feature of somatosensory plasticity stand out: the transition from LTD to LTP for post-pre pairs ($\Delta t = -10$ ms) at low firing rates between 5 and 15 Hz and consequentially the loss of distinction between pre-post and post-pre pair induced plasticity at $\sim 15$ Hz. In contrast for the visual cortex, LTD is induced for post-pre ($\Delta t = -10$ ms) stimulation from 0.1 up to about 25 Hz and the difference between pre-post and post-pre stimulation vanishes at 38 Hz. As a result, the range of different plasticity results for pre-post vs. post-pre stimulation is restricted to lower frequencies in the somatosensory- compared to the visual cortex.

The calcium traces for 10 and 22 Hz regular spike-pair stimulation in both structures (Fig 2E and 2F) demonstrate the two factors responsible for the transition from LTD to LTP at low firing rates in somatosensory cortex: (i) The difference between depression and potentiation threshold is small such that the LTP threshold is crossed and LTP is induced as soon as the calcium trace starts to accumulate for increasing frequencies ($\theta_d = 1$, $\theta_p = 1.388$ in somatosensory ctx.; $\theta_d = 1$, $\theta_p = 1.631$ in visual ctx.). (ii) The calcium decay time constant, $\tau_{Ca}$, is such that

consecutive calcium transients start to accumulate between 10 and 20 Hz stimulation ($\tau_{Ca} =$ 49.0 ms for somatosensory and 38.3 ms for visual cortex; see Table 1). In the visual cortex, the difference between both thresholds is larger and the calcium decay time constant is faster (see Table 1). As a result, 10 *and* 22 Hz stimulation dominantly activate depression leading to LTD for both cases.

Calcium responses to presynaptic stimulation suppress stronger in the somatosensory cortex (see Fig 1). In turn, any difference in the calcium trace between pre-post and post-pre stimulation disappears at lower stimulation frequencies in the somatosensory- compared to the visual cortex. This fact explains why the distinction in induced plasticity between pre-post and post-pre stimulation disappears at low stimulation frequencies in the somatosensory cortex. In other words, if the calcium traces for $\Delta t = -10$ ms and +10, 5 ms are alike, the times spent above potentiation and depression thresholds and therefore the induced plasticity are identical. This becomes furthermore apparent in the fit of the model variant without STD to the somatosensory plasticity data. Here, the difference in induced plasticity for pre-post vs. post-pre pairs is retained for frequencies up to $\sim 25$ Hz (see Fig 5*B*).

In summary, the spike-timing- and frequency dependence of synaptic plasticity can be captured by the calcium-based model endowed with STD dynamics in the calcium. In particular, the model resolves the experimentally measured difference between pre-post and post-pre spike-pair stimulation.

## Irregular spike-pair presentations strongly reduces the impact of spike-timing on synaptic plasticity

Spike-pairs in the plasticity experiments considered above were presented in a regular fashion, that is, with fixed inter-pair intervals. In a step towards more natural, irregular firing patterns, we use the previously suggested protocol of Poisson-distributed spike-pairs [22], where the presynaptic neuron fires spikes according to a Poisson process and each spike is followed by a postsynaptic spike at time lag $\Delta t$ with probability *p*. *p* effectively controls the strength of spiking correlation between pre- and postsynaptic neuron, with maximal correlation for *p* = 1 and independent Poisson firing for *p* = 0 (see Materials and methods).

A simple shift from regular pre- and postsynaptic spike-pairs to spike-pairs with the same timing constraints but irregular distribution has a strong impact on STDP curves (Fig 3). At all frequencies, the change in synaptic strength is dominated by LTP, while the LTD part of the curves is strongly reduced or even disappears for plasticity in the visual cortex (Fig 3*A*). Strikingly, the range of plasticity values obtained when varying the time lag between pre- and postsynaptic spikes, $\Delta t$, is much reduced for irregular spike pairs.

Irregular presentation of spike-pairs strongly reduced the impact of spike-timing on synaptic changes, which has been previously found for calcium- and triplet-based plasticity models [22]. This effect still holds true when STD is present in the calcium dynamics. Note that up to this point the reduction of the impact of spike-timing on plasticity for irregular spike-pairs is a prediction based on modeling work and remains to be tested experimentally.

## Synaptic changes occur at very different firing rate ranges across cortices

Next, we studied how synaptic changes in response to firing rate increases compare with changes from correlations and ask in particular in which firing rate range plasticity is most sensitive to both changes.

In visual- and somatosensory cortex, when increasing the uncorrelated firing rate in pre- and postsynaptic neurons, the change in synaptic strength follows a BCM-type of curve [40]:

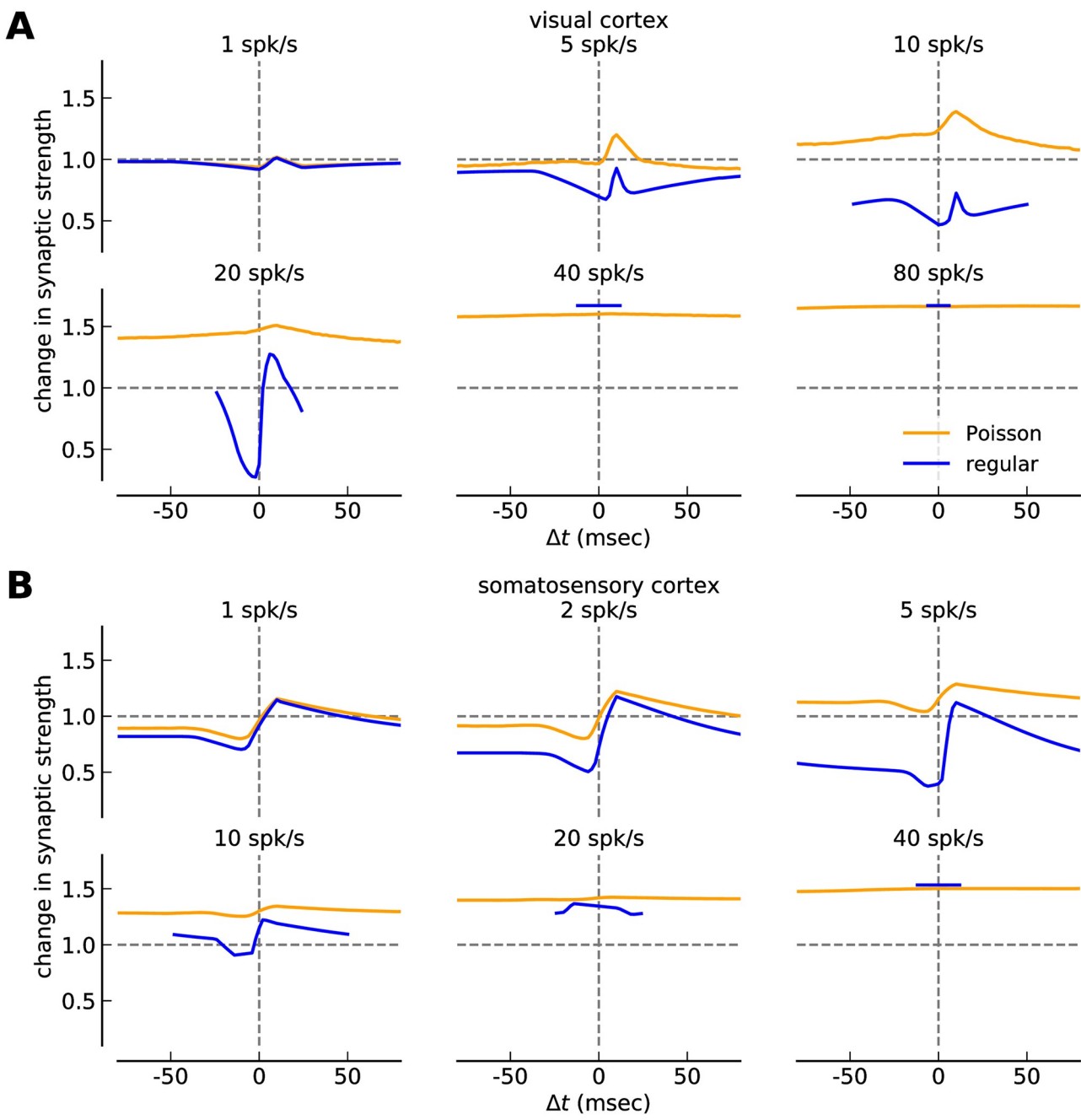

**Fig 3. Change in synaptic strength in response to irregular and regular spike-pairs at different firing rates.** *(A, B)* Change in synaptic strength as a function of the time lag between pre- and postsynaptic spikes for irregular (orange) and regular spike-pairs (blue) at different firing rates for the calcium-based plasticity model with STD dynamics of the calcium. The STDP curves for the visual cortex parameter-set is shown in *A* while somatosensory examples are shown in *B*. Note that all synaptic changes are shown from a 10 s stimulation and with *p* = 1, *i.e.*, the number of spikes occurring during a stimulation protocol varies with the firing rate and each presynaptic spike is followed or preceded by a postsynaptic one at time lag Δ*t*. Note that the curves for regular spike-pairs are shown for the interval Δ*t* ∈ [−1/(2ν), 1/(2ν)] only as the same curve is repeated for larger values of Δ*t*.

no change when both neurons are inactive, depression at low rates and potentiation at intermediate to high firing rates (Fig 4A and 4B).

Adding pre-post correlations (Δ*t* > 0 ms) increases the change in synaptic strength at low firing rates (*i.e.* < 25 spk/s in visual cortex and < 8 spk/s in somatosensory cortex; Fig 4C and 4D).

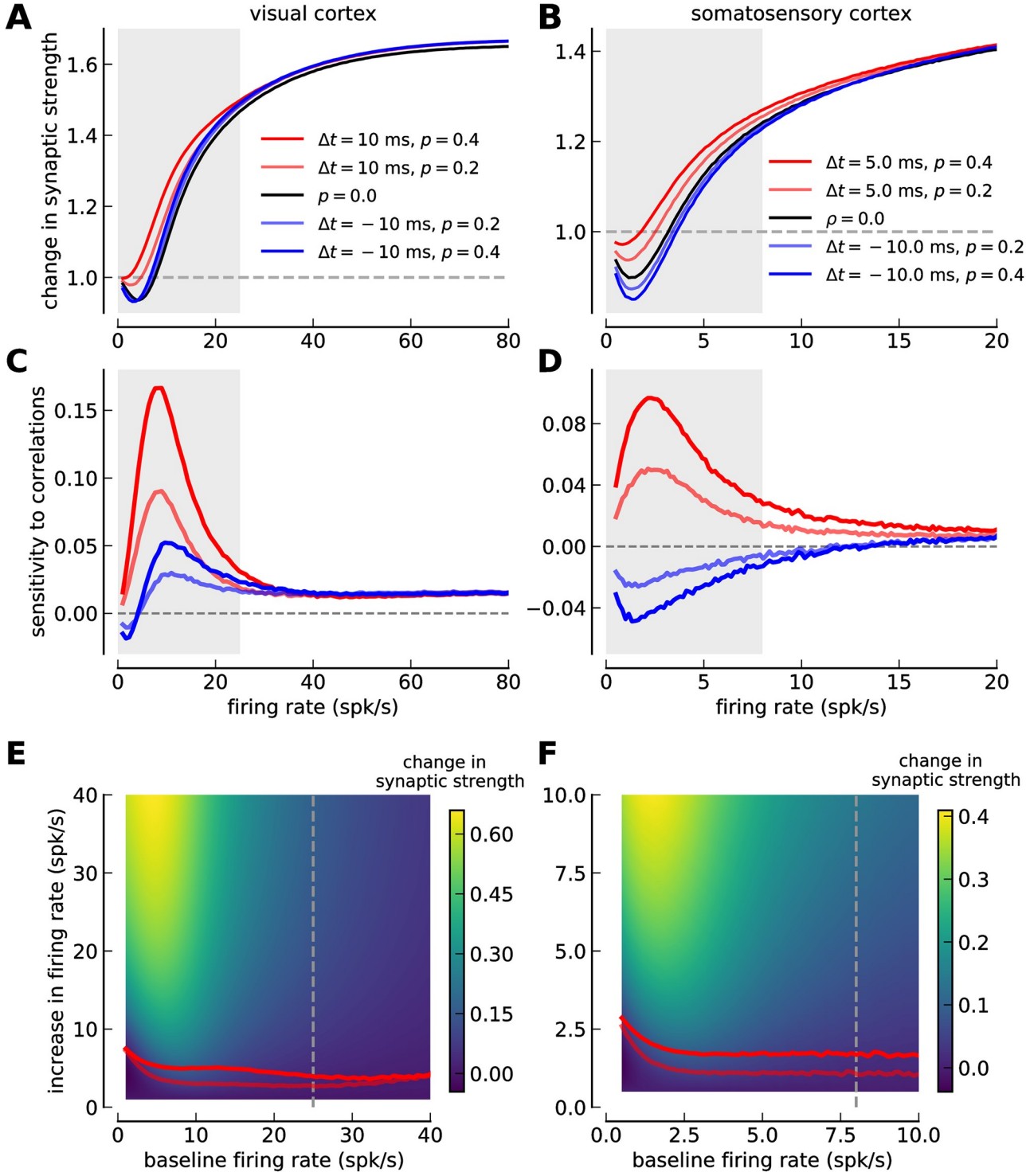

**Fig 4. Comparing the sensitivity of synaptic plasticity to firing rate and spike-timing between visual- and somatosensory cortex.** *(A, B)* Change in synaptic strength as a function of the firing rate for several values of the correlation coefficient $p$ and time lag $\Delta t$ in visual- *(A)* and somatosensory cortex *(B)*. Five cases are shown: (i) uncorrelated Poisson spike trains ($p = 0$, black), (ii) pre-post pairs with $\Delta = 10$ ms for visual cortex *(A)* and $\Delta = 5$ ms for somatosensory cortex *(B)* at $p = 0.2$ (light red) and $p = 0.4$ (dark red); and (iii) post-pre pairs with $\Delta = -10$ ms at $p = 0.2$ (light blue) and $p = 0.4$ (dark blue). $p = 0.2$ (0.4) implies that only 20% (40%) of presynaptic spikes are followed by a postsynaptic spike at delay $\Delta t$. *(C, D)* Sensitivity of synaptic changes to spike-pair correlations (see Materials and methods, Eq (8)) in visual- *(C)* and somatosensory cortex *(D)*. The change in synaptic strength due to spike-timing correlations is shown as a function of the firing rate. *(E, F)* Sensitivity of synaptic changes to firing rate changes (see Materials and methods, Eq (9)) in visual- *(C)* and somatosensory cortex *(D)*. Color plots represent the synaptic change as a function of the baseline firing rate (x-axis)

and the increase in firing rate (y-axis). Light and dark red lines indicate the firing rate increase evoking the same synaptic change as spike-pair correlations at $\Delta t = 10$ ms in *E* and $\Delta t = 5$ ms in *F* with $p = 0.2$ and $p = 0.4$, respectively. Prevalent firing rate ranges in visual ($< 25$ spk/s; [41]) and somatosensory cortex ($< 8$ spk/s; [42]) are marked by gray shaded regions in panels *A-D* and by a gray, dashed line in panels *E,F*. Note the different firing rate ranges shown on the x-axis between the left and the right column. All changes are in response to a stimulation for 10 s.

Post-pre correlations entail little change compared to uncorrelated pre- and postsynaptic activity in visual cortex and a decrease of the change in synaptic strength in somatosensory cortex (see blue lines in Fig 4C and 4D).

Even though the qualitative plasticity behavior is very similar between visual- and somatosensory cortex, that is, the sensitivity to firing rate and correlations exhibits the same overall behavior, the synaptic changes occur in very different firing rate ranges (note the different x-axis scales between the left and the right column in Fig 4). Synaptic plasticity is most sensitive to changes in firing rate and correlations for rates up to $\sim 15$ spk/s in the visual cortex, while the same amplitudes of sensitivities extends only up to $\sim 5$ spk/s in the somatosensory cortex (Fig 4). In other words, changes due to activity alterations are induced at much lower firing rates in the somatosensory cortex compared to the visual cortex. Similar to the difference in the sensitive ranges of plasticity, the prevalent firing rates in the visual- and somatosensory cortex are very different: while neurons in the visual cortex *in vivo* reach activities up to 25 spk/s [41], they fire up to 8 spk/s in the somatosensory cortex *in vivo* [42] (see gray shaded regions in Fig 4).

Where does this difference in sensitive ranges between somatosensory- and visual cortex come from? As discussed above, the transition from LTD to LTP at low firing rates in the somatosensory cortex emerges from the short extend of the LTD region, that is, the small difference between depression and potentiation thresholds (see Table 1). This difference is larger in the visual cortex which in turn gives rise to a transition to maximal LTP at higher firing rates. The dissimilar sensitivities to firing rate changes between visual- and somatosensory cortex are the result of this difference in potentiation thresholds.

Comparing model variants with and without STD fitted to the experimental data (Fig 5) illustrates that STD in the somatosensory cortex is responsible for restricting the correlation sensitivity to low firing rates $< 5$ spk/s. The somatosensory model variant without STD exhibits sensitivity to correlations up to $\sim 20$ spk/s which goes well beyond the prevalent firing rates in that region (see Fig 5F). With STD, the stronger suppression of presynaptically evoked calcium traces in the somatosensory cortex as compared to the visual cortex explains the disappearance of correlation sensitivity at low firing rates (see explanation above).

In summary, there is a perfect match between the predominant firing rates and the sensitive ranges of the synaptic plasticity in visual- and somatosensory cortex. Synaptic long- and short-term plasticity are tuned such that LTD and LTP occur at realistic firing rate ranges in both cortices. Fitting the calcium-based plasticity model to experimental plasticity data leads us to predict that the potentiation threshold—the calcium sensitivity of signaling cascades leading to LTP—is higher in the visual cortex compared to the somatosensory cortex. This difference is a crucial component creating the area specific sensitivity ranges.

## Calcium-based model with nonlinear calcium dynamics

In the calcium-based model, synaptic changes are driven by presynaptically and postsynaptically evoked calcium influx. Up to this point, we considered a simplified model of the calcium dynamics in which calcium transients elicited by presynaptic and postsynaptic spikes sum linearly. Could a more realistic, nonlinear calcium dynamics, mediated by the pre- and postsynaptic coincidence-dependent NMDAR current, change the sensitivity range of plasticity to spike

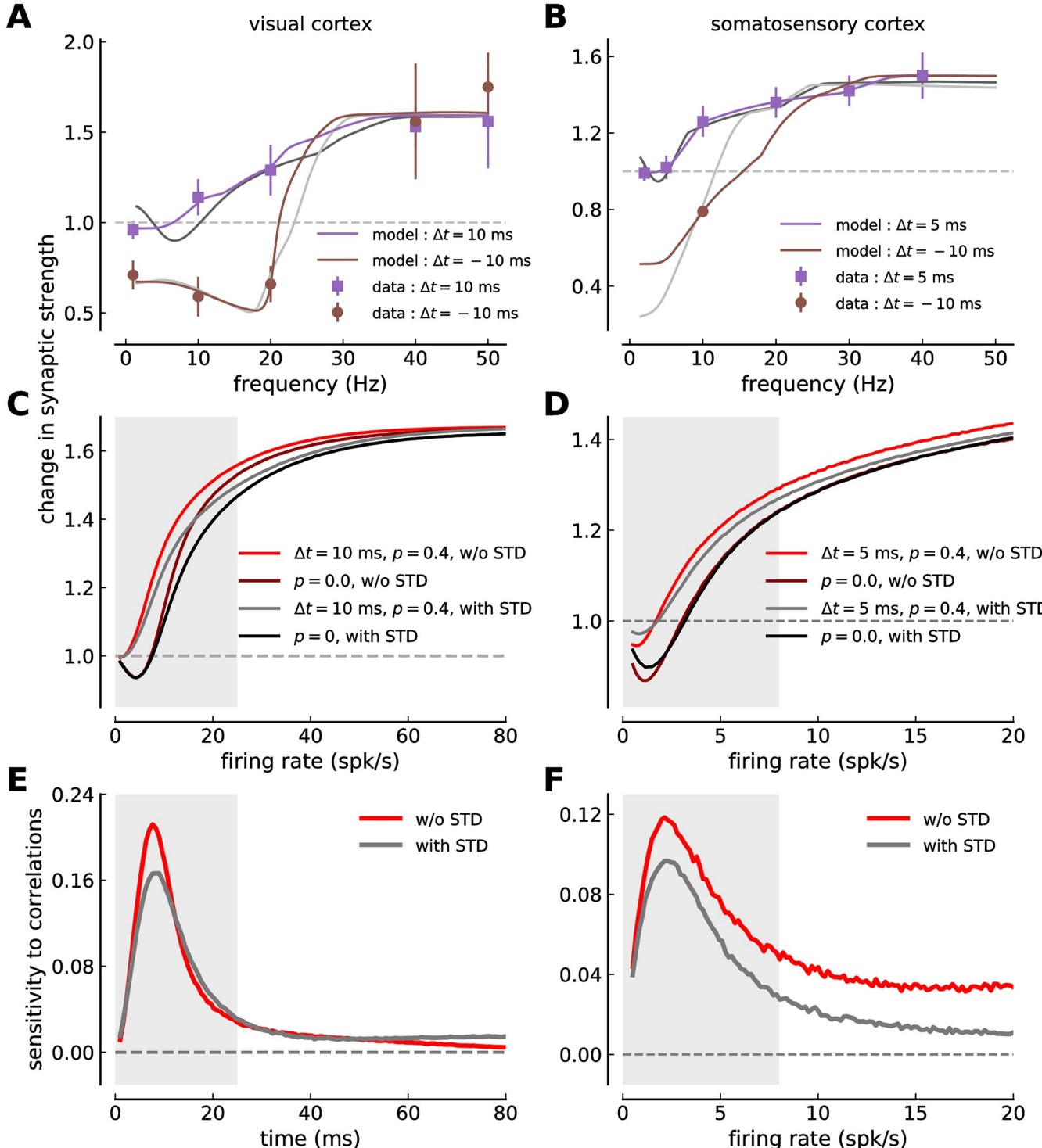

**Fig 5. Comparison between model versions with and without short-term depression.** *(A, B)* Model fit (solid lines) to the experimental plasticity data obtained in visual cortex (*A*, mean ± SEM, [2]) and somatosensory cortex slices (*B*, mean ± SEM, [3]). Same depiction as in Fig 2. The purple and brown lines show the model without STD fitted to the data, while the gray lines are a reproduction from Fig 2A and 2B of the model with STD. *(C, D)* Change in synaptic strength in response to irregular pre- and post activity as a function of the firing rate for several values of the correlation coefficient *p* and one time lag Δ*t* in visual- (*C*) and somatosensory cortex (*D*). Two cases are shown for the model without STD: (i) uncorrelated Poisson spike trains (*p* = 0, dark red) and (ii) pre-post pairs with Δ = 10 ms in *C* (Δ = 5 ms in *D*) at *p* = 0.4 (red). These two cases are also shown for the model with STD: (i) uncorrelated Poisson spike trains shown in black, and (ii) pre-post pairs with Δ = 10 ms in *C* (Δ = 5 ms in *D*) at *p* = 0.4 shown in gray. *(E, F)* Sensitivity of synaptic changes to spike-pair correlations in visual- (*E*) and somatosensory cortex (*F*). The change in synaptic strength due to spike-timing correlations is shown as a function of the firing

rate for the model without STD (red) and with STD (gray; same line as in Fig 4C and 4D). Prevalent firing rate ranges in visual (< 25 spk/s; [41]) and somatosensory cortex (< 8 spk/s; [42]) are marked by gray shaded regions in panels *C-F*. Note the different firing rate ranges shown on the x-axis between the left and the right column of panels *C-F*. All changes in panels *C-F* are in response to a stimulation for 10 s.

correlations and firing rate changes? To answer this question, we examined a nonlinear calcium dynamics implementation that includes a coincidence-dependent NMDAR component (Eq (6)). Here we show that our main results do not depend on the implementation of the calcium dynamics.

In the nonlinear calcium implementation, the calcium transient elicited by a postsynaptic spike consists of two components (see Materials and methods, Eq (6)): (1) a voltage-dependent calcium channel-mediated part; and (2) a nonlinear NMDA part controlled by a parameter that characterizes the increase of the NMDA mediated current in case of coincident presynaptic activation and postsynaptic depolarization, the nonlinearity factor $n$ (see Eqs (6) and (7)). The calcium transient elicited by a presynaptic spike remains unchanged and implements STD dynamics (Eq (3)).

To compare the results of the calcium-based model from the "nonlinear calcium dynamics" with the "linear calcium dynamics", we first fit the calcium-based model to visual- and somatosensory cortex plasticity data using the nonlinear calcium dynamics (Fig 6A and 6B). The calcium-based model with the nonlinear calcium implementation provides a good fit of the regular spike-pairs presented at different frequencies compared with the calcium-based model with linear calcium dynamics (compare colored and black/gray lines in Fig 6A and 6B).

We then compared the STDP curves for regular- and irregular spike-pairs at an intermediate rate varying the time lag between pre- and postsynaptic spikes (Fig 6 C,D). Note that the STDP curves exhibit a large jump with the nonlinear calcium model at the point at which the order between pre- and postsynaptic spike changes, *i.e.* at the point at which the nonlinear term (Eq (6)) is activated due to the pre-before-post order of spikes. When comparing synaptic changes induced by irregular spike-pairs with those induced by regular spike-pairs: (1) irregular spike-pairs induce less depression and more potentiation than regular spike-pairs; and (2) the influence of spike timing is reduced for irregular spike-pairs compared with regular spike-pairs. These differences between regular and irregular spike-pairs already observed with the linear calcium model seem to be further enhanced with nonlinear calcium dynamics (compare Fig 6C and 6D and Fig 3). Note also the large LTD magnitude for post-pre pairs ($\Delta t < 0$ ms) and regular spike-pair stimulation, which is a result of the permanently decreasing presynaptically induced calcium amplitudes due to STD and the scaling with the diminishing synaptic weight during the 10 s stimulation.

Lastly, we investigated synaptic changes in response to correlations for the nonlinear calcium model Fig 6E–6H. We asked in particular in which firing rate ranges plasticity is most sensitive in visual- and somatosensory cortex. The sensitivity to correlations reaches a peak at low firing rates (~10 spk/s in visual cortex; 4 spk/s in somatosensory cortex) and vanishes as the firing rates are increased further (Fig 6G and 6H). The results for the nonlinear calcium model are therefore qualitatively similar to the calcium-based model with linear calcium dynamics (compare red and gray lines in Fig 6G and 6H). In particular, the peak of correlation sensitivity falls in the range of prevalent firing rates in both structures. Quantitatively, the magnitude of the sensitivity to correlations is larger in the nonlinear calcium model compared to the linear version in visual cortex further enhancing the effect of selective sensitivity.

In summary, the extension of the model by a nonlinear version of the calcium dynamics demonstrates that the conclusions drawn on the cortex-specific sensitivity range to spike timing correlations does not depend on the calcium implementation.

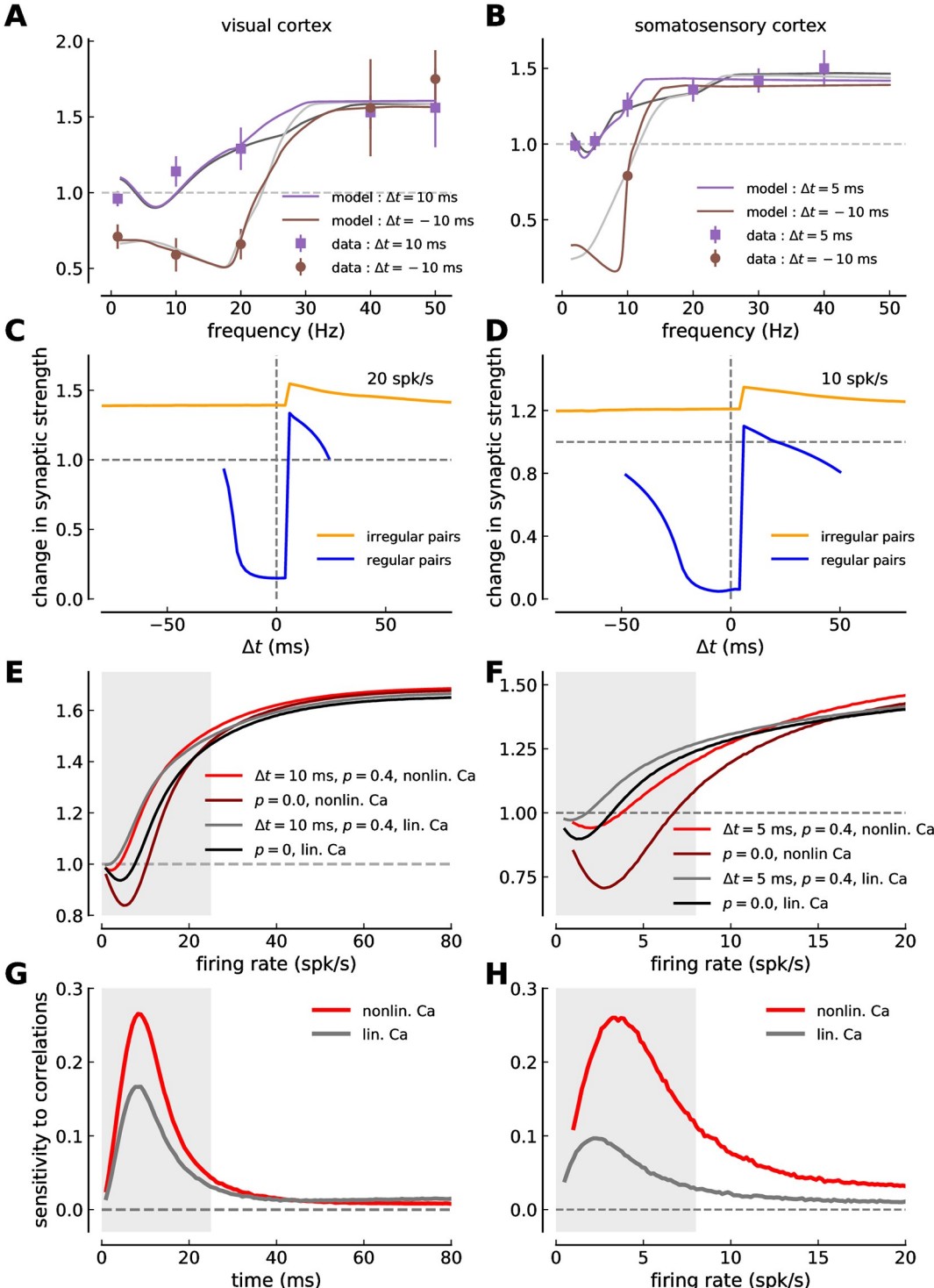

**Fig 6. Comparison of synaptic changes in linear- and nonlinear calcium dynamics model versions.** *(A, B)* Model fit (solid lines) to the experimental plasticity data obtained in visual cortex (*A*, mean ± SEM, [2]) and somatosensory cortex (*B*, mean ± SEM, [3]). Same depiction as in Fig 2. The purple and brown lines show the model with nonlinear calcium dynamics fitted to the data, while the gray lines are a reproduction from Fig 2A amd 2B of the model with linear calcium dynamics (see Materials and methods). Both model variants implement STD. *(C, D)* Change in synaptic strength as a function of the time lag between pre- and postsynaptic spikes for irregular (orange) and regular spike-pairs (blue) at an intermediate firing rate for the nonlinear calcium model. The STDP curve for the visual cortex parameter-set is shown in *C* while a somatosensory example is shown in *D*. *(E, F)* Change in synaptic strength in response to irregular pre- and post activity as a function of the firing rate for

several values of the correlation coefficient $p$ and one time lag $\Delta t$ in visual- (*E*) and somatosensory cortex (*F*). Two cases are shown for the model with nonlinear calcium dynamics: (i) uncorrelated Poisson spike trains ($p = 0$, dark red) and (ii) pre-post pairs with $\Delta = 10$ ms in *E* ($\Delta = 5$ ms in *F*) at $p = 0.4$ (red). These two cases are also shown for the linear calcium model: (i) uncorrelated Poisson spike trains shown in black, and (ii) pre-post pairs with $\Delta = 10$ ms in *E* ($\Delta = 5$ ms in *F*) at $p = 0.4$ shown in gray. *(G, H)* Sensitivity of synaptic changes to spike-pair correlations in visual- (*G*) and somatosensory cortex (*H*). The change in synaptic strength due to spike-timing correlations is shown as a function of the firing rate for the nonlinear calcium model (red) and for the linear model variant (gray; same line as in Fig 4C and 4D). Prevalent firing rate ranges in visual- ($<$ 25 spk/s; [41]) and somatosensory cortex ($<$ 8 spk/s; [42]) are marked by gray shaded regions in panels *E-H*. Note the different firing rate ranges shown on the x-axis between the left and the right column of panels *E-H*. All changes in panels *C-H* are in response to a stimulation for 10 s.

## Discussion

Using numerical simulations, we have explored the impact of short-term depression on long-term plasticity induction in visual- and somatosensory cortices. We fitted the calcium-based plasticity model to spike-pair- and frequency plasticity data in both structures and showed that the experimental data can be captured despite the activity-dependent reduction in presynaptically induced calcium transients during the burst stimulation and the scaling of evoked calcium transients with the dynamically changing synaptic weight. When examining plasticity in response to more *in vivo*-like, irregular stimulation patterns, we show that short-term and long-term plasticity parameters ensure that synaptic changes are susceptible to rate and correlation changes within the prevalent firing rates in both cortical areas, which are markedly different. Our findings suggest that long- and short-term synaptic plasticity are together tuned to account in combination for the activity properties of the synapse's location.

### Long-term plasticity alters short-term plasticity dynamics

The induction of long-term plasticity is known to alter short-term plasticity when long-term changes are expressed presynaptically [37, 43]. LTD induction, for example, reduces short-term depression due to a reduction in transmitter release. As a first universal approach, we use constant short-term plasticity parameters fitted to baseline responses between layer V pyramidal cells, that is, before long-term plasticity induction. At the same time, long-term changes affect presynaptically induced calcium amplitudes through an assumed linear scaling reflecting increases and decreases in synaptic strength in the evoked calcium response. See [32] for a phenomenological model which accounts for the change in short-term synaptic plasticity through long-term plasticity induction.

### Stochastic vesicle release

The deterministic short-term plasticity model utilized here is fitted to average postsynaptic responses. However, transmitter release is a stochastic process and as a consequence the magnitude of the postsynaptic response evoked by each presynaptic action potential fluctuates, even for the same preceding activity pattern. The quantal nature of synaptic transmission is described by binomial statistics [8, 36]. In turn, stochastic synaptic short-term plasticity parameters not only describe the change in the average response, but also the magnitude of fluctuations of individual responses. The variability of calcium responses impacts the induction of long-term plasticity in protocols repeating the same stimulation pattern multiple times (10 to 15 times as used here). It has been shown that a calcium-dependent plasticity model incorporating stochastic vesicle release better approximates experimental plasticity data of spike-triplets and is less sensitive to precise parameter values [29]. Whether experimental plasticity data considered here can be captured under such conditions and how stochasticity affects firing rate and correlation sensitivities is the subject of future research.

## Experimental predictions

To our knowledge, no experimental studies have explored how short-term plasticity shapes long-term plasticity during the induction phase. A likely reason for that is the interlinkage between both platicity processes which makes it experimentally difficult to alter one without affecting the other. Our study suggests that the strength of short-term depression controls the discrimination between pre-post and post-pre long-term plasticity when increasing the stimulation frequency (see Fig 5), *i.e.*, both stimulation patterns induce the same magnitude of LTP when pre-synaptic responses are strongly suppressed as it is the case in the somatosensory cortex. As a consequence, reducing experimentally short-term depression should reveal different long-term plasticity outcomes for post-pre and pre-post stimulation at elevated stimulation frequencies, as seen in the visual cortex at intermediate frequencies ($\sim$ 20 Hz) which exhibits weaker short-term depression compared to the somatosensory cortex. A possible way to reduced short-term depression experimentally is the depletion of the pool of release-ready vesicles with trains of presynaptic stimuli right before the long-term induction activity patterns, or by reducing the pause (10 s in [2] and 4 s in [3]) between the burst of spike-pairs, or by using genetic manipulations partly turning short-term depression into facilitation [44]. Our results predict a larger separation between pre-post and post-pre long-term plasticity in such cases. Another way to test our conclusions is to apply the same plasticity induction protocol in another system with markedly different short-term dynamics. The Schaffer collateral to hippocampal CA1 pyramidal neuron synapse would be an ideal candidate as it shows little short-term depression even at high stimulation frequencies [45]. A further prediction from our modeling study is the non-monotonic induction of LTP for pre-post ($\Delta t > 0$ ms) pairs. Due to short-term depression, the amount of induced LTP is reduced when increasing the stimulation frequency from 1 Hz to 5 Hz in both cortical areas considered (Fig 2*A* and 2*B*).

Recent theoretical studies highlighted the fact that Hebbian long-term plasticity alone can lead to unstable feedback loops in which correlations or high firing rates of pre- and postsynaptic neurons drive potentiation of synapses that further increase postsynaptic rates and correlations (see [46] for a review). Fast compensatory/homeostatic forms of synaptic plasticity such as fast heterosynaptic plasticity [47] or fast BCM-like metaplasticity [48] have been suggested to stabilize firing rates and Hebbian plasticity. The here proposed model combining short- and long-term plasticity effectively implements a fast compensatory mechanism for presynaptic stimulation through the reduction of presynaptically induced calcium transients due to STD dynamics. This is equivalent to an increase in the potentiation threshold—reminiscent of a sliding threshold—and can prevent runaway potentiation. Equivalently, LTD magnitude can be limited if calcium transients fall below the LTD threshold due to continued decrease of presynaptically evoked calcium transients. Whether these mechanisms are sufficient to assure stability and learning in neural circuits remains to be studied for the calcium-based model (see [33]), also since plasticity induced through postsynaptic activity is not compensated in the model.

## Validity of the results *in vivo*

Long-term plasticity in slices has been induced with elevated extracellular calcium concentrations while *in vivo* calcium levels are estimated to be around 1.5 mM [49]. Considering realistic calcium levels will most likely change the plasticity rules observed *in vitro* under elevated extracellular calcium [39] and has to be considered when applying our results to *in vivo* data. Most of the data on short-term plasticity stems from brain slices but a recent study established STD to be involved in the adaptation to sensory responses in the somatosensory cortex *in vivo* [50].

### Generality of model results

The deterministic short-term depression model utilized here has been fitted to evoked voltage responses between layer V neurons in somatosensory- and visual cortex. The two parameter sets describing STD dynamics are well constrained by these traces and provide a reliable account of the mean postsynaptic response, which is used for the calcium dynamics description here. Conversely, the calcium-based long-term plasticity model (with 8 free parameters) is insufficiently constrained by the 10 and 7 LTP/LTD data-points from visual- and somatosensory cortex (Fig 2*A* and 2*B*). The particular shape of the STDP curves is subject to this uncertainty. However, the frequency dependence of the plasticity is dominantly dictated by the calcium time constant, $\tau_{\mathrm{Ca}}$, which describes the interaction time scale between consecutive spikes. This parameter is well constrained by the regular plasticity data, and in turn, the main conclusions drawn here which concern the behavior of the model for interactions between consecutive stimuli are robust. Short-term depression models describing the use and recovery of presynaptic resources are typically applied to postsynaptic current responses [28, 29, 36]. Using them to directly model evoked calcium transients neglects the integration of current into concentration. How this simplification affects the conclusions in particular at high frequencies is the subject of future research.

Numerous experimental studies revealed a large diversity of long-term plasticity induction and expression mechanisms across different synapses. These studies have identified two key elements for the induction of long-term synaptic plasticity in hippocampus and neocortex. First, postsynaptic calcium entry mediated by N-methyl-D-aspartate receptors (NMDARs) [51] and voltage-dependent $Ca^{2+}$ channels (VDCCs) [38, 52, 53] has been shown in many cases to be a necessary [38, 54, 55] and sufficient [56–58] signal for the induction of synaptic plasticity. Second, calcium in turn triggers downstream signaling cascades involving protein kinases (mediating LTP) and phosphatases (mediating LTD) (see e.g. [59–62]). Another G-protein coupled LTD induction pathway, identified in visual- and somatosensory cortex, involves retrograde signaling by endocannabinoids which requires postsynaptic calcium elevations [37, 38, 63]. The highly simplified model used here retains the postsynaptic calcium signal as a crucial trigger of plasticity. The behavior of the model in response to spike-timing- and rate dependence during regular and irregular stimulation patterns arises from the interplay between postsynaptic calcium dynamics and the depression and potentation thresholds which implement in a highly simplified fashion calcium-dependent signaling cascades leading to synaptic potentiation and depression, respectively. How other plasticity induction pathways act in conjunction with calcium has been investigated for example in [64], but in what way this shapes plasticity with irregular stimulation remains to be studied.

### Conclusion

By including short-term plasticity effects in a calcium-based model of long-term plasticity, we aimed here to link the different time scales at which synapses modify their strength. Both, short-term and long-term plasticity influence each other and we suggest that this interaction might be region and synapse specific. Cortical cells have a large repertoire to adapt their responses to activity- or stimulus statistics on a range of time scales such as spike-frequency adaption on the millisecond scale [65], or homeostatic plasticity on the scale of hours [66]. Our results lend support to the idea that these mechanisms might act in a concerted fashion and that their dynamic ranges are well adjusted.

## Acknowledgments

We are thankful to Srdjan Ostojic for carefully reading the manuscript and providing helpful comments. We are indebted to Federico Trigo for valuable input in the early stage of this project.

## Author Contributions

**Conceptualization:** Nicolas Deperrois, Michael Graupner.

**Formal analysis:** Nicolas Deperrois, Michael Graupner.

**Funding acquisition:** Michael Graupner.

**Writing – original draft:** Michael Graupner.

**Writing – review & editing:** Nicolas Deperrois, Michael Graupner.

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
