## [Decision Letter · Decision Letter 0]

8 Nov 2019

Dear Dr Graupner,

Thank you very much for submitting your manuscript 'Short-term depression and long-term plasticity together tune sensitive range of synaptic plasticity' for review by PLOS Computational Biology. Your manuscript has been fully evaluated by the PLOS Computational Biology editorial team and in this case also by independent peer reviewers. The reviewers appreciated the attention to an important problem, but raised some substantial concerns about the manuscript as it currently stands. While your manuscript cannot be accepted in its present form, we are willing to consider a revised version in which the issues raised by the reviewers have been adequately addressed. We cannot, of course, promise publication at that time.

Sincerely,

Hermann Cuntz

Associate Editor

PLOS Computational Biology

Samuel Gershman

Deputy Editor

PLOS Computational Biology

[LINK]

I am happy that we found great reviewers for your manuscript and they found the work interesting but they had complementary concerns regarding the plasticity model and the context with respect to evidence from biology. Please address carefully all criticism by the reviewers.

Reviewer's Responses to Questions

**Comments to the Authors:**

Reviewer #1: The aim of the study is to investigate the interaction of short-term depression and long-term plasticity in the induction of synaptic depression and potentiation in visual- and somatosensory cortex. The modeling results show that the effect of short-term depression on long-term synaptic modifications is pronounced in both cortical areas for irregular and correlated spiking activity patterns, and it should be taken into consideration in the modeling studies.

The scientific question addressed is important and timely, the model designed is elegant, the results are clearly described, the figures are of high quality, and conclusions are supported by the simulation results.

Major comments:

Page 6:

Induction of long-term depression is mediated by presynaptic NMDAr and CB1 receptor as a retrograde messenger in somatosensory cortex (Sjostrom et al., 2003). The model does not include these components but rather relies on the postsynaptic NMDAr instead. The arguments are missing if such simplification can be made. Are the conclusions on the spike-timing- and frequency dependence of synaptic plasticity, firing rate ranges of synaptic changes, differences in synaptic plasticity parameters between somatosensory- and visual cortex valid?

Page 4:

The joint effect of the pre- and postsynaptic neuronal activity on the calcium concentration is described as the sum of pre- and postsynaptically evoked calcium amplitudes in the differential equation of calcium concentration (Eq 3). However, NMDAr shows a nonlinear behaviour that depends on the postsynaptic membrane potential, thus, on the temporal difference D. How will this nonlinearity affect the results and conclusions made?

Minor comments:

Page 3:

The term describing the dynamics of the synaptic efficacy in the absence of pre- and postsynaptic activity is missing in Eq 1. if compared to Graupner and Brunel (2012). Please make it clear why the model is chosen to be multistable, not bistable.

Page 5:

Why and how was the potentiation threshold chosen not to be optimized in the experiments for visual cortex? (Table 1)

Page 5:

Should be “The short-term plasticity parameters in the first two lines are obtained from ...“ instead of „The short-term plasticity parameters shown in the first two lines are obtain from... “ (Table 1)

Reviewer #2: In their computational study, the authors implemented short-term depression (STD) in a calcium-based model of LTP/LTD, which allowed them to study the interaction between short- and long-term synaptic plasticity. First, they tuned their model to STD electrophysiological data from the visual and the somatosensory cortex. STD was implemented as short-term plasticity of postsynaptic calcium responses during repeated presynaptic stimulation. Then the authors were able to tune the calcium-based long-term plasticity model (extended by STD) to replicate experimental LTP/LTD data from the visual and somatosensory cortex. They found that LTP and LTD in their STD/LTD/LTP model is induced at different firing rate ranges in the somatosensory and visual cortex, with higher firing rates in the visual cortex as compared to the somatosensory cortex. These different frequency ranges match the typical firing rates in the respective cortices suggesting that the interaction between short-term and long-term synaptic plasticity is tuned to increase the sensitivity of plastic changes of synapses to realistic firing rate ranges.

The paper addresses an important issue, since in most modelling and experimental studies, the interaction between short-term and long-term plasticity is neglected and this work contributes to our better understanding of it. The methods are well described and the results are appropriately interpreted and discussed. A strong aspect of the paper represent specific predictions and suggestions of experimental tests.

Major points:

* In the model, short-term plasticity affects postsynaptic calcium changes, which drive long-term synaptic changes, but these synaptic weight changes seem to be decoupled (?) from postsynaptic calcium responses. How would the results change if synaptic weight w affected Cpre (presynaptically evoked calcium amplitude)?

* A related point: The authors argue that due to separation of time scales (line 389), LTD/LTP can be decoupled from short-term depression. Their argument is that LTD/LTP induction protocols last from a few minutes to seconds while the expression of long-term plasticity takes tens of minutes. At least for LTP that does not seem to be the case. The expression of LTP (e.g. in the form of insertion of AMPARs) can be a fast process (e.g. upon high-frequency stimulation)

* When translating voltage events into calcium events, the authors assumed that “the evoked postsynaptic current is directly proportional to the induced calcium concentration”. Is this assumption based on spine calcium imaging data or is it a simplified assumption? Please explain in the paper.

* Has someone tested experimentally the important (previously published) prediction of the calcium-based model that irregular spike-pairs decrease the impact of spike-timing on synaptic changes? The authors should make clear whether this is still yet to be shown in experiments.

* Authors write that “dissimilar sensitivities to firing rate changes between visual- and somatosensory cortex are the result of [the] difference in potentiation thresholds.“ Is there experimental evidence for different plasticity thresholds in these two brain regions or is it a prediction of the model?

* Recent papers showed that fast compensatory/homeostatic (non-Hebbian) forms of synaptic plasticity such as fast heterosynaptic plasticity (Zenke et al. Nat Comm 2015) or fast BCM-like metaplasticity (Jedlicka et al. Plos CB 2015) on a time scale of seconds to tens of seconds can account for plasticity data and, importantly, stabilize firing rates and Hebbian plasticity (Zenke & Gerstner, Philos Trans R Soc Lond B Biol Sci. 2017). How could these rapid forms of homeostatic plasticity affect the interaction between short-term plasticity and long-term plasticity? It might be interesting to briefly discuss this in the Discussion.

-

Minor points:

Figure 1 legend: the text needs to distinguish between data and simulations. E.g. in Fig. 1B the authors could replace “Example calcium traces” e.g. by “Example simulations for calcium traces”. Fig.1A –are data points from somatosensory cortex missing? Based on what data was STD in the somatosensory cortex fitted?

Figure 4: To emphasize the differences in firing rate-dependency of synaptic plasticity between visual and somatosensory cortex, it would make sense to use the same scale (0 – 80 spk/s) for the x-axis.

Line 61: “To changes in synaptic efficacy under more natural conditions, we use

irregular spike patterns” should read “To simulate changes in…”

Line 315: “at time lag delta-t which probability p.” should read “at time lag delta-t with probability p.

Text for Table 1: In the sentence “The short-term plasticity parameters shown in the first two lines are obtain from fitting the STD model to”, “obtain” should read “obtained”

Fig. 4: The correlation variable p is not explained in the legend. Add the explanation. (See Fig 5 legend, in which it is explained).

Line 422: replace “reduced” by “reduce”

**Have all data underlying the figures and results presented in the manuscript been provided?**

Reviewer #1: Yes

Reviewer #2: Yes

PLOS authors have the option to publish the peer review history of their article (what does this mean?). If published, this will include your full peer review and any attached files.

Reviewer #1: No

Reviewer #2: No

---

## [Decision Letter · Decision Letter 1]

17 Aug 2020

Dear Dr. Graupner,

We are pleased to inform you that your manuscript 'Short-term depression and long-term plasticity together tune sensitive range of synaptic plasticity' has been provisionally accepted for publication in PLOS Computational Biology.

Best regards,

Hermann Cuntz

Associate Editor

PLOS Computational Biology

Samuel Gershman

Deputy Editor

PLOS Computational Biology

Reviewer's Responses to Questions

**Comments to the Authors:**

Reviewer #1: The authors have addressed all the comments carefully and in detail, and the manuscript can be accepted for publication.

It is strongly recommended to make the model available on ModelDB (https://senselab.med.yale.edu/modeldb/).

Reviewer #2: The authors revised the manuscript in line with all my major points.

**Have all data underlying the figures and results presented in the manuscript been provided?**

Reviewer #1: Yes

Reviewer #2: Yes

PLOS authors have the option to publish the peer review history of their article (what does this mean?). If published, this will include your full peer review and any attached files.

Reviewer #1: No

Reviewer #2: No

---

## [Editor Report · Acceptance letter]

22 Sep 2020

PCOMPBIOL-D-19-01244R1 

Short-term depression and long-term plasticity together tune sensitive range of synaptic plasticity

Dear Dr Graupner,

I am pleased to inform you that your manuscript has been formally accepted for publication in PLOS Computational Biology. Your manuscript is now with our production department and you will be notified of the publication date in due course.

With kind regards,

Matt Lyles
